# Factors Influencing the Decision to Vaccinate against HPV amongst a Population of Female Health Students

**DOI:** 10.3390/vaccines10050680

**Published:** 2022-04-25

**Authors:** Laure Nicolet, Manuela Viviano, Cheryl Dickson, Emilien Jeannot

**Affiliations:** 1Institute of Global Health, Faculty of Medicine, Chemin de Mines 9, 1202 Geneva, Switzerland; laure.nicolet@unige.ch; 2Gynecology Division, Department of Obstetrics and Gynecology, Geneva University Hospitals, Boulevard de la Cluse 30, 1205 Geneva, Switzerland; manuela.viviano@hcuge.ch; 3Community Psychiatric Service, Lausanne University Hospital (CHUV), Rue du Bugnon 23, 1011 Lausanne, Switzerland; dr.cheryldickson@live.com

**Keywords:** human papillomavirus, vaccination, invasive cervical cancer, student population

## Abstract

Background: In Switzerland, HPV vaccination has been recommended since 2007 for all adolescent girls aged between 11 and 14 years. More than 10 years after the introduction of this recommendation, immunization coverage targets have not been met. Very few studies at a national level describe the reasons for the reluctance of some young women to become vaccinated. The aim of this study is to describe the socio-demographic characteristics of a population of vaccinated and unvaccinated female health students and then to compare the different factors that may have influenced their vaccine choice. Method: Female health students in the French-speaking part of Switzerland, aged between 18 and 31, were invited to participate in the study. A total of 234 female students completed a questionnaire that included questions about their socio-demographic data, sexual behavior and vaccination status. Results: 69% of the participants received at least one dose of the vaccine. Women who had not yet had sex were less likely to be vaccinated than sexually active women (ORa: 0.1, 0.0–0.4, 95% CI), the same as those who did not express an opinion about the importance of vaccination (ORa: 0.1: 0.0–0.6, 95% CI). The main reasons given for refusing vaccination were fear of side effects (26.0%), parental opposition (24.6%) and reluctance of the attending physician (13.6%). Conclusions: The main results of this study highlight a good rate of vaccine coverage in the sample population. Reasons for nonvaccination demonstrate the need to provide information on the vaccine to the target audience, as well as to parents and health professionals.

## 1. Introduction

Human papillomavirus (HPV) is a group of sexually transmitted viruses, of which there are more than a hundred genotypes. These viruses infect the skin and genital mucous membranes. Although most infections are self-resolving, some genotypes can cause genital warts (6 and 11) as well as precancerous lesions and invasive cancers [1]. Both types 16 and 18 are responsible for 70% of invasive cervical cancer [2]. The development of such cancer is preceded by precancerous lesions.

Globally, cervical cancer is the 4th most diagnosed cancer and the 4th leading cause of cancer-related death in women [3]. For the year 2020, 604,000 new cases and 342,000 deaths were recorded [3]. In Switzerland, around 5000 women a year are diagnosed with precervical cancer, leading to additional examinations and sometimes surgery. Around 260 types of cancer are diagnosed each year in Switzerland [4]. In terms of frequency, cervical cancer is ranked as the fifth most common type of cancer found among Swiss women aged 20 to 49 [5].

The primary prevention strategy to reduce the risk of invasive cervical cancer is to vaccinate young women, ideally before they are sexually active. The vaccinal goal is to effectively reduce the burden of disease by avoiding the onset of precancerous stages [6]. The development of vaccines against human papillomavirus has therefore represented a major advancement in prevention. A Swedish study demonstrated a significant reduction in the risk of invasive cervical cancer. Specifically, an 88% reduction was found for young women vaccinated before the age of 17 and 50% for those vaccinated between ages 17 and 30, when compared with an unvaccinated sample [7].

In Switzerland, vaccination has been recommended since 2007 to all adolescent girls aged between 11 and 14 years (before they become sexually active) [8]. It is also offered to adolescent girls aged 15 to 26 years as a catch-up vaccination. Since 2007, two vaccines protecting against oncogenic genotypes 16 and 18 have been available: Cervarix^®^ and Gardasil^®^ [9]. Gardasil^®^ also offers protection against genotypes 6 and 11 which are responsible for genital warts. The effectiveness of these two vaccines against invasive cervical cancers is estimated to be 70% [10]. Since 2019, a new vaccine has been used, Gardasil 9^®^, as it offers additional protection against five other oncogenic papillomavirus genotypes [11].

Since autumn 2008, all of the 26 Swiss cantons have implemented vaccination programs. The aim of these campaigns was to achieve vaccination coverage rates of 80% for girls aged 11 to 14 years and 50% amongst those aged 15 to 19 years [12]. A study published in 2019 shows an estimated vaccination coverage of 54% amongst 18- to 20-year-olds and 34% amongst 21- to 24-year-olds, with significant regional differences. These results indicate that six years after the start of the program these objectives had not yet been achieved [13]. Cantons in which vaccination is carried out through school programs have higher vaccination coverage rates than other cantons [14].

In the Swiss French-speaking cantons, 3 years after the start of the campaign, the vaccination coverage rate was 63% for vaccination with one or two doses and 61% with three doses [15]. This result, which is higher than the national average, is largely due to good coordination at a cantonal level. The involvement of child and adolescent health services has contributed to increasing vaccination coverage. However, this rate remains low compared with the 80% target initially set [12]. Very few studies at a national level describe the reasons for the reluctance of some young women to become vaccinated. In order to improve and further target prevention programs, it is essential to understand the individual factors influencing the choice to vaccinate. A study of 16 to 20 year olds living in French-speaking Swiss cantons revealed a perceived lack of information about the history of and prevention approaches towards HPV. In particular, over 70% of the target population reported feeling insufficiently informed about these issues. The lack of information about HPV and its vaccine was not restricted to Switzerland and affects many other countries.

Another study in Sweden has shown that HPV vaccination among young girls was associated with a European background and high maternal education level, as well as more favorable beliefs about HPV prevention and less sexual risk taking.

Research has identified that health professionals play an important role in vaccine uptake. Moreover, there is a lack of initiatives to improve education among undergraduate healthcare students about HPV infection, its associated consequences and methods of prevention.

The aim of this study is to describe the socio-demographic characteristics of a population of female health students in two Swiss French-speaking cantons (Vaud and Geneva). A comparison is then made between the vaccinated and unvaccinated groups to determine whether certain factors may influence a decision in favor of vaccination.

## 2. Method

### 2.1. Population and Context

This study was carried out between September 2019 and January 2020 in the cities of Geneva and Lausanne in Switzerland. The participants were female students of nursing, midwifery and dietetics (in years 1 to 3 of a bachelor’s degree). All students aged between 18 and 31 who agreed to participate in the study were included. Students were excluded if they had a history of hysterectomy or treatment targeting the cervix in the past 12 months. All individuals who fully completed the questionnaire and agreed to participate in the study were included in our sample.

### 2.2. Procedure of the Study

This study was conducted as a cross-sectional observational study. Information about the study was communicated to the students via the University of Geneva and University of Lausanne websites as well as by the investigators themselves via email. In addition, a presentation of the study was made to the participants during class hours.

At the end of the course, the questionnaire was distributed to students interested in participating in the study. Participants completed the questionnaire at home. The completed questionnaires were returned to the investigators within one week of distribution. Alternatively, participants could complete the socio-demographic questionnaire online via a web platform. The survey instrument was an online self-administrated, anonymous questionnaire developed using SurveyMonkey software (https://de.surveymonkey.com/, accessed on 1 July 2019). This software automatically saves responses to a secure database, thus protecting the participants’ confidentiality. On the first page of the web questionnaire, the participants were presented with a consent form informing them of the objectives of the study and its procedures. The students had the right to refuse or terminate their participation in the study at any moment, in which case the time of study drop-out was indicated in the questionnaire. If they agreed to participate in the study, they were asked to tick a box in order to declare their informed consent. If the individuals did not agree, the webpage automatically closed. Three email reminders were sent at 1, 2 and 3 weeks after the first invitation, unless an individual requested to be removed from the mailing list during the process. The web-based survey was automatically closed 10 weeks after the first invitation was sent.

### 2.3. Data Collection

In this study, data obtained through a questionnaire were analyzed. All participants answered questions about their socio-demographic profile (age and nationality), sexual behavior (number of sexual relationships and condom use) and HPV vaccination status (number of doses received, age at first dose and opinion on the vaccination). The questionnaire used for data collection was developed and validated in a previous study using the same population type [16].

### 2.4. Statistical Analysis

Statistical analyses were carried out using STATA 14. Descriptive statistics and frequencies were analyzed for all variables with a 95% confidence interval. The Kolmogorov–Smirnov test was used to analyze the normality of distribution for the continuous variables. Normally distributed continuous variables were reported as means and standard deviations (SDs), and categorical variables were reported as frequencies (%). A ƿ value of at least 0.05 was considered to be statistically significant. Multivariate logistic regression was performed to identify factors influencing vaccine choice. Vaccination status was used as the primary outcome. Only variables identified to be of interest in the univariate analysis were included in the multivariate analysis. The final model was selected with a stepwise procedure based on Akaikes information criterion, and the results were reported as odds ratios with 95% confidence intervals.

### 2.5. Ethical Approval

The study protocol was approved by the University Hospital of Geneva’s Cantonal Commission for Ethics and Research on Human Beings (approval number 19-357). This study was conducted in accordance with all laws and regulations relating to Good Practice in Clinical Trials (ICH E6-1996) and the Declaration of Helsinki (Fortaleza, Brazil, October 2013). All participants signed a consent form before taking part in the study.

## 3. Results

### 3.1. Sociodemographic Data

Data from the questionnaires of 234 female students was analyzed in this study. Characteristics of the participants are presented in Table 1.

Participants’ mean age at the start of the study was 21 years (range 17 to 29 years). The majority of participants were of foreign nationality (73%) compared with Swiss women, who represented 27% of the sample. Of the students surveyed, 78% reported consuming tobacco regularly, 14% sometimes and 8% never. The majority of participants (92%) had already had a first experience of sexual intercourse. The average age reported at the time of the first experience was 17 years (range 10 to 24 years). The average number of sexual partners reported by female students was 5.4.

In terms of condom use as a means of contraception, 28% of participants reported that they rarely or never used a condom. In total, 24% stated that they sometimes used one, 24% often used a condom and 23% always used one. A total of 8% of the females surveyed had never had sex and therefore did not answer the question about contraception.

Following vaginal self-sampling, 77% of participants were not carriers of a human papillomavirus. Four students (2%) were found to be carriers of a human papillomavirus type 6, 11, 16 or 18, and 51 students (22%) were carriers of another genotype of the virus.

### 3.2. Vaccination Status

Regarding the vaccination rate, 161 students (69%) had received at least one dose of HPV vaccine. Of these, 84% had received three doses, 9% two doses and 6% only one dose. All participants had been vaccinated with the Gardasil^®^ vaccine. The average age at the time of receiving the first dose was 14.6 years (range 9 to 26 years). A total of 52% of the participants consulted their vaccination record in order to answer the questionnaire.

Regarding the participants’ opinions on HPV vaccination, 71% thought that this vaccination was as important as other vaccinations, 16% thought it was less important, 3% thought it was more important and 9% had no opinion on the issue.

### 3.3. Association between HPV Vaccination and Sociodemographic Characteristics

Table 2 presents the association between the HPV vaccination and sociodemographic characteristics. The results show that young women who had not yet had sexual intercourse were significantly less vaccinated than sexually active young women (ORa: 0.1, 0.0–0.4 95% CI).

One particularly important finding is that participants who did not express an opinion about vaccination had a lower probability of being vaccinated (ORa: 0.1, 0.0–0.6 95% CI).

There was no statistically significant difference between the two groups in terms of nationality, tobacco consumption, number of sexual partners and HPV test positivity.

### 3.4. Reasons Given for Vaccination Refusal

Table 3 shows that of the 73 participants who had not been vaccinated, 62 had mentioned the reason for their refusal of the vaccine. These different reasons are presented in Table 3. The most common reason given for refusing vaccination was fear of side effects, reported by 26% of unvaccinated young women. Parental opposition (24%) was also among the most frequently cited reasons, alongside that of the attending physician (13%). The attending physicians were general practitioners, pediatricians or gynecologists.

Three other reasons were less frequently mentioned: namely the absence of sexual intercourse (4%), the absence of hindsight (2%) and lack of information (2%). The reasons categorized as “other” were extraneous to the analysis and included comments such as “I do not want to vaccinate myself” or “I never have vaccines”.

## 4. Discussion

In this study, 69% of participants received at least one dose of the HPV vaccine. Although lower than the initial vaccination target of 80% [12], this percentage is quite acceptable in comparison with the existing vaccination coverage rate among 18 to 20 yearolds in Switzerland, which is estimated to be 54% [13]. A study published in 2011 reported a vaccination rate of 64% for a single dose amongst 11- to 19-year-old girls in the canton of Geneva [15]. This good coverage rate can be explained, in particular, by a cantonal campaign aimed at easier access to vaccinations. The campaign offers teenagers the opportunity to be vaccinated as part of compulsory schooling. Several studies have already shown that countries with school-based information programs have higher vaccination coverage rates [17,18]. In the present study, the good vaccination coverage rate cannot be linked solely to the efforts of the cantonal campaign as a high proportion of the participants are not Swiss.

The average age at the time of injection of the first dose was 14.6 years. This is in line with the recommendations that vaccination be given before the age of 15 [8]. The age of the participants at the time of vaccination suggests that they were vaccinated in a school setting or by their pediatrician.

Regarding factors that may influence participants’ decision to vaccinate, a significant association could be found for the onset of sexual activity. Indeed, young women who had not yet had sexual intercourse were 10 times less likely to be vaccinated than those who had already had intercourse. This result suggests that women who do not have sex feel less concerned by vaccination. However, analysis of the reasons given for vaccine refusal indicates that only 4% of unvaccinated women cite not yet having had sex as the main reason. Their vaccine refusal does not therefore seem to be related to this single factor.

Evidence from the literature suggests mixed results concerning the relationship between vaccination and the onset of sexual activity. One Swedish study shows that a high number of sexual partners as well as an onset of sexual activity at a young age are predictors for vaccination [19]. However, the results of the Swiss study by Wymann et al. show that women who have had more than 10 sexual partners are less often vaccinated than those with only 1 or 2 [13]. These contradictory results demonstrate the need to continue efforts to inform adolescents of the importance of vaccination before the onset of sexual activity.

The other statistically significant variable in this analysis is participants’ opinions about the importance of vaccination. Women without an opinion are 10 times less likely to be vaccinated than those who reported an opinion on the issue. This lack of position from a significant proportion of female students is surprising and could suggest a lack of information or interest in the HPV vaccination. The link between lack of knowledge about the benefits of vaccination and vaccine refusal has already been studied. Particularly, an Italian study by Restivo et al. shows that women with less knowledge of the HPV vaccination are less likely to be vaccinated than others [20].

In relation to vaccination uptake, 85% of unvaccinated participants cited a reason for refusing the vaccine. The three main reasons given are the fear of side effects, opposition from parents or that of the attending physician. The reasons given in the literature depend very much on the country and the psychosocial context. One systematic review published in 2017 highlights parents’ views and lack of information as the main barriers to vaccination [21]. However, it notes the importance of physician recommendations and parental acceptance as facilitating factors. Other studies also put the source of information as a key element in encouraging vaccination [22]. Young women with unofficial (nonprofessional) sources of information, such as friends for example, have a higher probability of not getting vaccinated [23].

### Strengths and Limitations of the Study

The key strength of this study is that it analyses the data of future health professionals. The participants of this study will probably have to give information to their future patients, and it is therefore essential that they are made aware of issues relating to HPV and vaccination.

However, certain limitations should be taken into account when interpreting the results. Firstly, as this study had a cross-sectional design, we could not determine a causal relationship but only hypotheses and reflections about factors that might encourage or discourage vaccination. Secondly, this sample concerns a very specific population, mostly foreign health students in two cities, and is therefore not representative of the general Swiss population. Another limitation is that our sample is relatively small, resulting in low statistical power. Thus, we cannot know whether a larger sample would have allowed us to find statistically significant associations between the socioeconomic variables and decision on whether to vaccinate.

In order to have a more precise and complete idea of the influence of factors facilitating vaccination, the use of a standardized tool such as the CHIAS (Caroline HPV Immunization Attitudes Scale) [24] could be considered for data collection.

## 5. Conclusions

The main results of this study show that in this population of undergraduate healthcare students a good level of HPV vaccination coverage has been achieved, even if it does not yet reach the level desired by the health authorities. The results demonstrate the need to provide information on vaccination. Specifically, if we increase the knowledge and awareness level of the target audience, this should reduce the reluctance for HPV vaccinations and increase the coverage rate. The reasons given for nonvaccination demonstrate that public health programs should also include health professionals and parents in order to achieve the objectives set. The results of this study, as well as the existing literature, indicate the need for further large studies with longitudinal follow-up, ideally using mixed and qualitative methods. These would enable confirmation of the link between various factors and the decision to vaccinate, in order to provide prevention programs that promote vaccination. Additional studies could also be carried out to evaluate the effectiveness of interventions encouraging vaccination among young Swiss women.

## Figures and Tables

**Table 1 vaccines-10-00680-t001:** Sociodemographic characteristics.

Characteristic	Sample Population (*n* = 234)		
	*n* *	% or Mean	95% CI
**Age**			
Mean age	218	21.0	20.7–21.3
**Nationality**			
Swiss	63	26.9%	21.6–33.0
Other	171	73.1%	67.0–78.4
**Tobacco consumption**			
Never	19	8.1%	5.2–12.4
Sometimes	33	14.1%	10.2–19.2
Often	182	77.8%	72.0–82.7
**Already had first sexual intercourse**			
Yes	214	91.5%	87.1–94.4
No	20	8.6%	5.6–12.9
**Age at first sexual intercourse**			
Mean age	213	17.0	16.7–17.3
**Number of sexual partners**			
Mean number	234	5.4	4.5–6.2
**Condom use as contraception**			
Never/rarely	60	28.0%	22.4–34.5
Sometimes	52	24.3%	19.0–30.5
Often	52	24.3%	19.0–30.5
Always	50	23.4%	18.1–27.6
**HPV carrier**			
No	179	76.5%	70.6–81.5
Yes (types 6, 11, 16, 18)	4	1.7%	0.6–4.5
Yes (other type)	51	21.8%	16.9–27.6
**Vaccinated against HPV**			
Yes	161	68.8%	62.5–74.4
No	73	31.2%	25.6–37.5
**Vaccine doses received**			
1	10	6.2%	3.4–11.2
2	15	9.3%	5.7–14.9
3	135	83.9%	77.3–88.8
Do not know	1	6.0%	0.1–4.3
**Age at first dose**			
Mean age	158	14.6	14.2–15.0
**Consultation of vaccination record**			
Yes	120	51.5%	45.1–57.9
No	113	48.5%	42.1–54.9
**Opinion about HPV vaccination**			
More important than others	7	3.0%	1.4–6.2
Less important than others	38	16.2%	21.0–21.6
As important as others	167	71.4%	65.2–76.8
Do not know	22	9.4%	6.3–13.9

* The value of n varies between the variables based on the number of no-responses and questions addressed only to a subgroup of participants.

**Table 2 vaccines-10-00680-t002:** Association between vaccination and sociodemographic characteristics: results of univariate and multivariate analyses.

Sociodemographic Characteristics	OR	95% CI	Adjusted OR	95% CI
**Nationality**				
Swiss	1.0	-	1.0	-
Other	0.6	0.3–1.2	0.5	0.2–1.1
**Tobacco consumption**				
Never	1.0	-	1.0	-
Sometimes	0.4	0.1–1.5	0.5	0.1–2.3
Often	0.9	0.3–2.5	0.8	0.2–3.2
**Already had first sexual intercourse**				
Yes	1.0	-	1.0	-
No	0.4	0.2–1.1	**0.1**	**0.0–0.4**
**Number of sexual partners**				
1	1.0	-	1.0	-
2 to 5	1.0	0.5–2.1	1.4	0.6–3.3
>5	0.9	0.4–2.0	2.1	0.7–5.8
**Opinion about HPV vaccination**				
More important than others	1.0	-	1.0	-
Less important than others	0.4	0.1–2.3	0.2	0.0–1.7
As important as others	1.5	0.3–8.1	1.2	0.2–7.3
Do not know	**0.1**	**0.0–0.8**	**0.1**	**0.0–0.6**
**HPV carrier**				
No	1.0	-	1.0	-
Yes (types 6, 11, 16, 18)	0.4	0.6–3.0	0.1	0.0–1.2
Yes (other type)	0.7	0.4–1.3	0.5	0.2–1.1

Statistically significant results in **bold**. The odds ratios were adjusted according to the following variables: nationality, tobacco consumption, already had first sexual intercourse, number of sexual partners and opinions about HPV vaccination and HPV carrier.

**Table 3 vaccines-10-00680-t003:** Reasons given for refusal of vaccination.

Reason	Sample Population (*n* = 73)	
	*N*	%
No reason given	11	15.1
Reason given *n* = 62:		
Fear of side-effects	19	26.0
Parental opposition	18	24.6
Physician opposition	10	13.6
Never had sexual intercourse	3	4.1
Absence of hindsight	2	2.7
Lack of information	2	2.7
Other	8	10.9

## Data Availability

Data will be made available by the authors upon reasonable request.

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
