# Peer review of "Factors Influencing the Decision to Vaccinate against HPV amongst a Population of Female Health Students"

_vaccines, 2022, doi:10.3390/vaccines10050680_

Round 1

Reviewer 1 Report

The authors present a study of sociodemographic characteristics and reasons for not taking the HPV vaccination among female health students in Switzerland. Overall, the study is well presented and the statistical methods sound.

A major limitation is the small sample size (n=234), implying lack of power and high probability of type II errors. As there was a lack of association between most of the sociodemographic variables and vaccination status, this could be due to the low power of the study.

In the introduction, there should be some references to other studies on this topic (sociodemographic characteristics and reasons behind HPV vaccination reluctance).

In the intro, there also should have been a motivation for the choice of study population (female health students).

In the conclusions, the sentence: "The mixed results from the literature demonstrate the need for longitudinal studies to confirm the link between certain factors and the decision to vaccinate, in order to be able to best target programs to promote vaccination." is hard to understand without some additional explanation.

In table 1 and table 2, tobacco is mis-spelled as tabacco.

Author Response

Referee: 1

Comment 1 :    The authors present a study of sociodemographic characteristics and reasons for not taking the HPV vaccination among female health students in Switzerland. Overall, the study is well presented and the statistical methods sound.

Answer 1 :       Thank you very much for your interest in this article and for the opportunity  to improve it by responding to your comments.

Comment 2 :    « A major limitation is the small sample size (n=234), implying lack of power and high probability of type II errors. As there was a lack of association between most of the sociodemographic variables and vaccination status, this could be due to the low power of the study»

Answer 2 :       We agree with you that this is a limitation of the study and we have therefore stated this in the paragraph entitled strengths and limitations of the study, in the discussion section.

Comment 3 :    In the introduction, there should be some references to other studies on this topic (sociodemographic characteristics and reasons behind HPV vaccination reluctance).

Answer 3:        We have added a paragraph in the introduction section to provide more references to other studies sociodemographic characteristics and reasons behind HPV vaccination reluctance.

Comment 3 :    In the intro, there also should have been a motivation for the choice of study population (female health students).

Answer 3:        We have added a paragraph in the introduction section to provide more details about our choice of this specific study population.

Comment 3 :    In the conclusions, the sentence: "The mixed results from the literature demonstrate the need for longitudinal studies to confirm the link between certain factors and the decision to vaccinate, in order to be able to best target programs to promote vaccination." is hard to understand without some additional explanation

Answer 3:        We have modified this concluding sentence to make it more understandable.

Comment 4 :    In table 1 and table 2, tobacco is mis-spelled as tabacco

Answer 4:  Indeed it is an error we have changed the term in tables 1 and 2. 

Reviewer 2 Report

Thanks to the authors for this interesting manuscript. It deals with the definition of the factors influencing the decision to vaccinate against HPV amongst a population of 234 female health students aged between 18-24 in Switzerland.

This manuscript gives an interesting viewpoint about vaccination hesitancy and globally the conclusions fit with prior research related to vaccination adherence and hesitancy.

My only major concern related to the readability of the results.
I suggest to the authors add some kind of figure showing the leading attributes of the decision to vaccinate against HPV.

Some minor English proofreading can be done.

Author Response

Referee: 2

Comment 1:    « Thanks to the authors for this interesting manuscript. It deals with the definition of the factors influencing the decision to vaccinate against HPV amongst a population of 234 female health students aged between 18-24 in Switzerland.

This manuscript gives an interesting viewpoint about vaccination hesitancy and globally the conclusions fit with prior research related to vaccination adherence and hesitancy.»

Answer 1 :       Thank you very much for your positive feedback and helpful suggestions. . We have done our best to update the paper, in keeping with your comments.

Comment 2:    « My only major concern related to the readability of the results.
I suggest to the authors add some kind of figure showing the leading attributes of the decision to vaccinate against HPV.»

Answer 2 :       Thank you for your comment and suggestion. Having considered this point, we believe that the results are better explained here by using a table rather than a graph. We have therefore adjusted the body of the text to provide a more detailed explanation of the results.

Comment 3:    «Some minor English proofreading can be done. »

Answer 3 :       The whole article has now been proofread by an English-speaking editor and the grammar has been revised where necessary.

Reviewer 3 Report

Please find my general comments below.

It is not clear if the questionnaire used is valid and reliable.

The sampling method has many issues, and there is no logical reason why only 234 students were selected.

More information on the data collection process is necessary.

It is not clear at all how the socio-demographic profile was assessed.

Which Multivariate logistic regression model was applied and how?

First, smoking status: firstly, smoking consumption is not a correct phrase, and secondly, it is not clear how this variable was assessed.

Table 3, Lack of information is provided as a reason for vaccine refusal. This is not a reason. The authors were not aware of missing data.

Author Response

Referee: 3

Comment 1:    It is not clear if the questionnaire used is valid and reliable.>

Answer 1 :       We have explained in the method section where the questionnaire came from and why we consider it valid.

Comment 2:    « The sampling method has many issues, and there is no logical reason why only 234 students were selected.»

Answer 2 :       We have explained in the method section how the sample was constituted.

Comment 3: More information on the data collection process is necessary

Answer 3 : We have added further information in the method section on how the data was collected.

Comment 4 : It is not clear at all how the socio-demographic profile was assessed.

Answer 4 :       In the method section we tried to explain better how the variables were evaluated.

Comment 5 : Which Multivariate logistic regression model was applied and how?

Answer 5 : In the statistical analysis section we have added information on the type of multivariate analysis performed.

Comment 6 : First, smoking status: firstly, smoking consumption is not a correct phrase, and secondly, it is not clear how this variable was assessed.

Answer 6 : For the evaluation of this variable, we have explained in the method section how it was evaluated using a previously validated questionnaire, we also invite the reader to see the primary article that used this questionnaire.

We have changed the term smoking consumption to tobacco consumption which is indeed more correct

Comment 7 : Table 3, Lack of information is provided as a reason for vaccine refusal. This is not a reason. The authors were not aware of missing data.

Answer 7 : Thank you for your comment. Could you please let us know what you would like us to change in table 3 to address the referee's comments?

Round 2

Reviewer 1 Report

There are a few places where two words are written as one (e.g wouldhave in line 274). Please go through the paper to eliminate these typos. 

I would have change the sentence: "Another limitation is that our sample is relatively small and so we may have lacked a Beta statistic power." to "Another limitation is that our sample is relatively small, resulting in low statistical power."

Author Response

Thank you for your second set of comments. We have tried to address the issue you have raised, as far as possible.

Comment 1 :    There are a few places where two words are written as one (e.g wouldhave in line 274). Please go through the paper to eliminate these typos. 

Answer 1 :       We have proofread the text and corrected these typos

Comment 2 :    « I would have change the sentence: "Another limitation is that our sample is relatively small and so we may have lacked a Beta statistic power." to "Another limitation is that our sample is relatively small, resulting in low statistical power."»

Answer 2 :       We have modified this sentence, as per your suggestion.

Reviewer 3 Report

The following is a poor conclusion for this study. The main results of this study highlight a good rate of vaccine coverage in the sample population. Reasons for non-vaccination demonstrate the need to provide information on the vaccine to the target audience, as well as parents and health professionals.

I would suggest a practical conclusion.

Also, there are many spelling and grammatical errors.

The study design is cross-sectional with relatively a very low sample size. So, please don't emphasize the associations, because these associations found are not causal.

Still, the sampling method and the reasons for including this small sample size are not convincing.

Author Response

Referee: 3

Thank you for your second set of comments. We have tried to address the issue you have raised, as far as possible.

Comment 1:    The following is a poor conclusion for this study. The main results of this study highlight a good rate of vaccine coverage in the sample population. Reasons for non-vaccination demonstrate the need to provide information on the vaccine to the target audience, as well as parents and health professionals.

Answer 1 :       We have modified the conclusion of the article to better highlight the main results that you have cited.

Comment 2:    Also, there are many spelling and grammatical errors.

Answer 2 :       The latest version of this article has now been proofread by an English-speaking editor in order to correct the spelling and grammatical errors.

Comment 3:  The study design is cross-sectional with relatively a very low sample size. So, please don't emphasize the associations, because these associations found are not causal.

Answer 3 : We are  agree with this comment regarding the non-causal aspect of the study and have therefore explained this in the section entitled strengths and limitations of the study.

Comment 4 : Still, the sampling method and the reasons for including this small sample size are not convincing.

Answer 4 :       We are sorry that our argument on sampling did not convince you. We are open to any suggestions to improve this part. We admit that we expected to have a higher participation rate with a more substantial N. Nevertheless this 1st study, even if incomplete, has enabled us to secure funding for a follow-on study with a longitudinal design and a larger sample size.

Round 3

Reviewer 3 Report

[No more comments!]